



# Large watershed flood forecasting with high resolution
# distributed hydrological model
Yangbo Chen[1], Ji Li[1], Huanyu Wang[1], Jianming Qin[1], Liming Dong[1]
[1]Department of Water Resources and Environment, Sun Yat-sen University,
Guangzhou 510275, China
*Correspondence to:* Yangbo Chen (eescyb@mail.sysu.edu.cn)
**Abstract:** Flooding is one of the most devastating natural disasters in the world with
huge damages, and flood forecasting is one of the flood mitigation measurements.
Watershed hydrological model is the major tool for flood forecasting, although the
lumped watershed hydrological model is still the most widely used model, the
distributed hydrological model has the potential to improve watershed flood
forecasting capability. Distributed hydrological model has been successfully used in
small watershed flood forecasting, but there are still challenges for the application in
large watershed, one of them is the model's spatial resolution effect. To cope with this
challenge, two efforts could be made, one is to improve the model's computation
efficiency in large watershed, another is implementing the model on high performance
supercomputer. By employing Liuxihe Model, a physically based distributed
hydrological model, this study sets up a distributed hydrological model for the flood
forecasting of Liujiang River Basin in southern China that is a large watershed.
Terrain data including DEM, soil type and land use type are downloaded from the
website freely, and the model structure with a high resolution of 200m*200m grid cell
is set up. The initial model parameters are derived from the terrain property data, and
then optimized by using the PSO algorithm, the model is used to simulate 29 observed
flood events. It has been found that by dividing the river channels into virtual channel
sections and assuming the cross section shapes as trapezoid, the Liuxihe Model
largely increases computation efficiency while keeping good model performance, thus
making it applicable in larger watersheds. This study also finds that parameter





uncertainty exists for physically deriving model parameters, and parameter
optimization could reduce this uncertainty, and is highly recommended. Computation
time needed for running a distributed hydrological model increases exponentially at a
power of 2, not linearly with the increasing of model spatial resolution, and the
200m*200m model resolution is proposed for modeling Liujiang River Basin flood
with Liuxihe Model in this study. To keep the model with an acceptable performance,
minimum model spatial resolution is needed. The suggested threshold model spatial
resolution for modeling Liujiang River Basin flood is 500m*500m grid cell, but the
model spatial resolution at 200m*200m grid cell is recommended in this study to keep
the model a better performance.
**Key words**:watershed flood forecasting, distributed hydrological model, Liuxihe
Model, parameter optimization, model spatial resolution

## 1 Introduction

Flooding is one of the most devastating natural disasters in the world, and huge
damages has been caused (Krzmm, 1992, Kuniyoshi, 1992, Chen, 1995, EEA, 2010).
Flood forecasting is one of the most widely used flood mitigation measurements, and
watershed hydrological model is the major tool for flood forecasting. Currently the
most popular hydrological model for watershed flood forecasting is still the so-called
lumped model (Refsgaard et. al., 1996), which averages the terrain property and
precipitation over the watershed, so do the model parameters. Hundreds of lumped
models have been proposed and widely used, such as the Sacramento model proposed
by Burnash et. al. (1995), the Tank model proposed by Sugawara et. al. (1995), the
Xinanjiang model proposed by Zhao (1977), and the ARNO model proposed by
Todini (1996), only naming a few among others. It is widely accepted that the
precipitation for driving the watershed hydrological processes is usually unevenly
distributed over the watershed, particularly for the large watershed, so the lumped
model could not easily forecast the watershed flooding of large watersheds.



Furthermore, due to the inhomogeneity of terrain property over the watershed, which
is true even in very small watershed, so the watershed flood forecasting could not be
forecasted accurately if the model parameters are averaged over the watershed. For
this reasons, new models are needed to improve the watershed flood forecasting
capability, particularly for large watershed flood forecasting.

Development of distributed hydrological model in the past decades provides the
potential to improve watershed flood forecasting capability. One of the most
important features of the distributed hydrological model is that it divides watershed
terrain into grid cells, which are regarded to have the same meaning of a real
watershed, i.e., the grid cells have their own terrain properties and precipitation. The
hydrological processes are calculated at both the grid cell scale and the watershed
scale, and the parameters used to calculate hydrological processes are also different at
different grid cells. This feature makes it could describe the inhomogeneity of both
the terrain property and precipitation over watershed. The distributed feature of the
distributed hydrological model is a very important feature compared to lumped model,
which makes it could better simulate the watershed hydrological processes at all scale,
small or large. The inhomogeneity of precipitation over watershed could also be well
described in the model, this is very helpful in modeling large watershed hydrological
processes, particularly in the tropical and sub-tropical regions where the flooding is
driven by heavy storm. For this reason, distributed hydrological model is usually
regarded to have the potential to better simulate or forecast the watershed flood
(Ambroise et. al., 1996, Chen et. al., 2016). Employing distributed hydrological
model for watershed food forecasting has been a new trend(Vieux et. al., 2004, Chen
et. al., 2012, Céline Cattoën et. al., 2016, Witold et. al., 2016, Kauffeldt et. al., 2016).

The blueprint of distributed hydrological model is regarded to be proposed by Freeze
and Harlan (1969), the first distributed hydrological model was the SHE model
proposed by Abbott et. al. (1986a, 1986b). Distributed hydrological model requires





different terrain property data for every grid cells to set up the model structure, so it is
data driven model. In the early stage of distributed hydrological modeling, this posted
great challenge for distributed hydrological model's application as the data was not
widely available and inexpensively accessible. With the development of remote
sensing sensors and techniques, terrain data covering global range with high
resolution has got readily available and could be acquired inexpensively. For example,
the DEM at 30m grid cell resolution with global coverage could be freely downloaded
(Falorni et al., 2005, Sharma et. al., 2014), which largely pushes forward the
development and application of the distributed hydrological models. After that, many
distributed hydrological models have been proposed, such as the WATERFLOOD
model (Kouwen, 1988), THALES model (Grayson et al., 1992), VIC model (Liang et.
al., 1994), DHSVM model (Wigmosta et. al., 1994), CASC2D model (Julien et. al.,
1995), WetSpa model (Wang et. al., 1997), GBHM model (Yang et. al., 1997), WEP-L
model (Jia et. al., 2001), Vflo model (Vieux et. al., 2002), tRIBS model(Vivoni et. al.,
2004), WEHY model (Kavvas et al., 2004), Liuxihe model (Chen et. al., 2011, 2016),
and more.

Distributed hydrological model derives model parameters physically from the terrain
property data, and is regarded not need to calibrate model parameter, so it could be
used in data poor or ungauged basins. This feature of distributed hydrological model
made it applied widely in evaluating the impacts of climate changes and urbanization
on hydrology(Li et. al., 2009, Seth et. al., 2001, Ott, et. al., 2004, Vanrheenen et. al.,
2005, Olivera et. al., 2007). But it also was found that this feature caused parameter
uncertainty due to the lack of experiences and references in physically deriving model
parameters from the terrain property, so could not be used in fields that require high
flood simulated accuracy, including watershed flood forecasting. It was realized that
parameter optimization for distributed hydrological model is also needed to improve
the model's performance, and a few methods for optimizing parameters of distributed
hydrological model have been proposed. For example, Vieux et. al. (2003) tried a




so-called scalar method to adjust the model parameters, and the model performance is
found to be improved largely. Madsen et. al. (2003) proposed an automatic
multi-objective parameter optimization method with SCE algorithm for SHE model,
which improved the model performance also. Shafii et. al. (2009) proposed a
multi-objective genetic algorithm for optimizing parameters of WetSpa model, the
improved model result is regarded to be reasonable. Xu et. al. (2012) proposed an
automated parameter optimization method with SCE-UA algorithm for Liuxihe Model,
which improved the model performance in a small watershed flood forecasting. Chen
et. al. (2016) proposed an automated parameter optimization method based on PSO
algorithm for Liuxihe Model watershed flood forecasting, and tested in two watershed,
one is small, one is large. The results suggested that distributed hydrological model
should optimize model parameters even if there is only little available hydrological
data, while the derived model parameters physically from the terrain perperty could
serve as an initial parameters. The above progresses in distributed hydrological
model's parameter optimization has matured, and will largely improve the
performance of distributed hydrologcial model, thus pushing forward the application
of distributed hydrologcial model in real-time watershed flood forecasting.

Spatial resolution is a key factor in distributed hydrological modeling. Theoretically if
the spatial resolution of a distributed hydrological model is higher, i.e., the grid cell
size is smaller, the terrain property could be described finer, and the hydrological
processes could be better simulated or forecasted, so the model spatial resolution
should be as high as possible. But on the other hand, higher model spatial resolution
requires higher resolution terrain property data for model setting up which may not be
available in some watersheds. But the most important is that distributed hydrological
model uses complex equations with physical meanings to calculate the hydrological
processes, so it needs much more computation resources than that of lumped model,
and the required computation resources increases exponentially with the increasing of
the model spatial resolution. So in modeling flood processes of a large watershed, the





computation time needed for running the distributed hydrological model will be huge
if the model spatial resolution is kept high, which may make the model application
impractical due to high running cost. So if distributed hydrological model is needed to
be applied in large watershed, a coarser resolution is the only choose, and the model's
capability will be impacted with less satisfactory results. This is also called the scaling
effect of distributed hydrological modeling. For this reason, current application for
watershed flood forecasting either limited to small watershed with higher resolution
or coarser resolution in large watershed, i.e., a trade-off between the model
performance and running cost.

Nowadays forecasting large watershed flooding has been in great demands as it
impacts peoples and their properties at large range, but due to the scale effect, current
distributed hydrological models employed for large watershed are at coarser
resolution, which lowers its capability for flood forecasting and warning. For example,
past application of distributed hydrological model for large watershed flood forecating
are at the resolution coarser than 1km grid cell (Lohmann et. al., 1998, Vieux et. al.,
2004, Stisen et. al., 2008, Rwetabula et. al., 2007), the models employed in the
pan-European Flood Awareness System (EFAS; Bartholmes et. al., 2009, Thielen et.
al., 2009, 2010, Sood et. al., 2015, Kauffeldt et. al., 2016) are at 1-10km grid cell,
which makes the result only applicapble for flood warning.

Challenge for distributed hydrological model application in large watershed flood
forecasting is its need for huge computation resources, to cope with this challenge,
two efforts could be made. One is to improve the computation efficiency of the
distributed hydrological modeling in large watershed, another is implementing the
model on high performance supercomputer so in the cases that the users are willing to
pay a high computation cost, the flood forecasting of large watershed with high
resolution could be done. In this study, the Liuxihe Model (Chen et. al., 2011, 2016), a
physically based distributed hydrological model proposed for watershed flood



forecasting, has been tried for flood forecasting of a large watershed in southern
China to validate the feasibility of distributed hydrological model's application for
large watershed flood forecasting.

## 2 Studied river basin and data

### 2.1 Liujiang River Basin

The river basin studied in this paper is the Liujiang River Basin(here after referred to
as LRB) in southern China, which is the first order tributary of the Pearl River. LRB
originates from Village Lang in Guizhou Province, and drains though Guizhou
Province, Guangxi Zhuang Autonomous Region and Hunan Province with 72% of its
drainage area in Guangxi Zhuang Autonomous Region. The length of its main channel
is 1121 km, the total drainage area is 58270 km$^2$ that marks it a large river basin in
China. Fig. 1 is a sketch map of LRB.

Fig. 1 sketch map of Liujiang River Basin(LRB)

LRB is a mountainous watershed in southern China. There are high mountains in the
north and northwest of the watershed with high elevation, while in its south and
southeast area, the elevation is low. This topography helps forming severe flooding in
the middle and downstream. The basin is in the sub-tropical monsoon climate zone
with an average annual precipitation of 1800 mm, and the precipitation distribution is
highly uneven both at spatial and temporal with 80% of its annual precipitation occurs
in the summer. LRB is in the center of storm zone of Zhuang Autonomous Region,
heavy storm was very frequent in the past. There are 59 disastrous flooding in the past
400 years with recording since 1488, which makes LRB the tributary with most
serious flooding among all the first order tributaries of the Pearl River. In the
watershed, there is no significant flood mitigation project to store flood runoff, so
flood forecasting is one of the most effective ways for the flood management.

### 2.2 Hydrological data

There are 66 rain gauges installed in the watershed. In this study, hydrological data of
30 flood events has been collected, including the precipitation of the rain gauges and





the river discharge of Liuzhou river gauge that locates in the downstream of the
watershed and closes to the outlet as shown in Fig. 1 with a hourly step, brief
information of these flood events is listed in Table 1.

Table1 Brief information of flood events with data collected in LRB

**2.3 Terrain property data**
Terrain property data includes DEM, land use/cover map and soil map, which are
used for setting up the distributed hydrological model for flood forecasting. In this
study, the DEM was downloaded from the SRTM database (Falorni et al., 2005,
Sharma et. al., 2014), the land use type was downloaded from the USGS land use type
database (Loveland et. al., 1991, Loveland et. al., 2000), and the soil type was
downloaded from FAO soil type database (http://www.isric.org). The downloaded
DEM has a spatial resolution of 90m*90m, considering LRB is large, the running load
for the model with a resolution of 90m*90m may be too heavy to run in this study, so
the DEM is rescaled to the resolutions of 200m*200m, as shown in Fig. 2(a). The
downloaded land use and soil type were at 1000m*1000m resolution, so there are
rescaled to the same resolution of DEM, as shown in Fig. 2(b) and Fig. 2 (c)
respectively.

Fig. 2 Terrain properties of LRB

The highest elevation and the lowest elevation of LRB are 2124 m and 42 m
respectively. There are 9 land use types, including evergreen needle leaved forest,
evergreen broadleaved forest, shrubbery, mountain and alpine meadow, slope
grassland, urban area, river, lakes and cultivated land, accounting for 18.1%, 31.0%,
32.5%, 0.1%, 13.7%, 0.1%, 0.2%, 0.3% and 4% of the total drainage area respectively.
Forestry, including evergreen needle leaved forest and evergreen broadleaved forest is
the major land use type with a percentage of 49.1%, shrubbery occupies a big portion
of the watershed also with a percentage of 32.5%, slope grassland also has a
significant portion with a percentage of 13.7%, other land use types are very less and
are not significant, this means LRB is well vegetated.




There are 11 soil types, including Humicacrisol, Haplic and high activitive acrisol,
Ferralic cambisol, Haplicluvisols, Dystric cambisol, Calcaric regosol, Dystric regosol,
Haplic and weak active acrisol, Artificial accumulated soil, Eutricregosols and Black
limestone soil, Dystric rankers, accounting for 0.8%, 1.5%, 5%, 3.5%, 2.8%, 45.5%,
2.9%, 18%, 1.5%, 3.5% and 15% of the total drainage area respectively. Calcaric
regosol is the major soil type which occupies 45.5% of the watershed area, almost half
of the drainage area, which is mainly in the east side of the watershed. Haplic and
weak active acrisol is another major soil type with an area percentage of 18% and is
located in the west side of the watershed. Dystric rankers is also a major soil type with
an area percentages of 15% which located in the north side of the watershed. Other
soil types are not significant with area percentages below 5% respectively and scatted
within the watershed.

## 244     3 Liuxihe Model for LRB flood forecasting

### 245     3.1 Introduction of Liuxihe Model

Liuxihe Model is a physically based distributed hydrological model proposed mainly
for watershed flood forecasting (Chen, 2009, Chen et. al., 2011, 2016). Like other
distributed hydrological models, Liuxihe Model divides the watershed into grid cell
based on the DEM of the studied watershed. To keep a reasonable model performance,
in the past experiences of Liuxihe Model research and application, the model
resolution is limited to 90m*90m or 100m*100m, but only used in small watersheds
(Chen, 2009, Chen et. al., 2011, 2013, 2016, Liao et. al., 2012 a, b, Xu et. al., 2012 a,
b). Precipitation, evaporation and runoff production are calculated at cell scale, runoff
routes first on cell, then alone the cell to river channel, and finally to the watershed
outlet. As Liuxihe model is mainly used in the sub-tropical regions, so the runoff
production is calculated based on the saturation-excess mechanism. The runoff
routing is classified as hill slope routing, river channel routing, subsurface routing and
underground routing. The hill slope routing is regarded as the one-dimensional
unsteady flow, and the kinematical wave approximation is employed to do the routing.





The river channel routing is also regarded as the one-dimensional unsteady flow, but
the diffusive wave approximation is employed to do the routing. The above methods
are widely used in the dominated distributed hydrological models.

What makes Liuxihe Model different is that the river channel cross section shape is
assumed to be trapezoid. With this assumption, the river channel size could be
represented with 3 indices, including the bottom width, side slope and bottom slope.
One of the advantages with this assumption is that the river channel cross section size
could be estimated with remotely sensed data, so Liuxihe Model could do river
channel runoff routing real physically, thus making Liuxihe Model a fully distributed
hydrological model. As there are too many river channel cross sections, and many of
them are in the upstream of the watershed where it is not easily accessed, so in real
hydrological modeling, directly measuring the river channel cross section sizes are
impractical. For this reason, most of the distributed hydrological model could not be
applied in real applications, or simply route the runoff with lumped methods which
makes the model not a fully distributed hydrological model, thus lowering the model's
capability in simulating or forecasting the watershed flood processes. Another
advantage of this assumption is that it also simplifies the runoff routing, thus
improves the model's computation efficiency. For this reason, even Liuxihe Model
has a very high resolution, it still could be used in real-time flood forecasting. This
feature of Liuxihe Model in estimating river channel cross section sizes makes it has
the potential to be used in large watershed flood forecasting.

Like other distributed hydrological model, when used in ungauged or data poor
watershed flood forecasting, Liuxihe Model derives model parameters physically
from the terrain property data, but automatic parameter optimization methods have
been tried, and two methods, including the SCE-UA algorithm (Xu et. al, 2012) and
PSO algorithm (Chen et. al., 2016) have been successfully used for Liuxihe Model's
parameter optimization. Study results also suggested that the parameter uncertainty is



289 high for the physically derived model parameters, and if there is a few observed

290 hydrological processes data, model parameter optimization is recommended that

291 could improves the model performance largely (Chen et. al., 2016). But as automatic

292 parameter optimization needs thousands model runs, that makes it difficult to be used

293 widely due to huge computing source requirement, which also make it taking long

294 time in setting up the model. For this reason, a public computer cloud was set up for

295 optimizing the parameters of Liuxihe Model which employs parallel computation

296 techniques and was implemented on a supercomputer system(Chen et. al., 2013). With

297 this development, Liuxihe Model could easily optimize its model parameters.

299 Above advancements of Liuxihe Model in estimating river channel cross section sizes

300 with remotely sensed data, automatic parameters optimization and supercomputing

301 makes it has the potential to be used in large watershed flood forecasting, so in this

302 study, the Liuxihe model is employed to study the LRB's flood forecasting.

303 **3.2 Liuxihe Model set up**

304 Considering LRB is large, so the DEM with 200m×200m resolution is adopted to set

305 up the model structure, not at the original 90m×90m resolution. The whole watershed

306 is first divided into 1469900 cells by the DEM horizontally, which were further

307 categorized into hill slope cells and river cells. By using Strahler method (Strahler,

308 1957), the river channel is divided into 3 order system as shown in Fig. 3, which

309 divides the whole cells into 1463204 hill slope cells and 6696 river cells.

310  Fig. 3 Liuxihe Model structure set up for LRB (200m×200m resolution)

311 To estimate the river channel sizes, 178 virtual nodes were set on the river channel

312 system, and 225 virtual channel sections were formed as shown in Figure 3. As in

313 Liuxihe Model, the shape of the virtual channel sections is assumed to be trapezoid,

314 so the cross section size is represented by three indices, including bottom width, side

315 slope and bottom slope. As proposed in Liuxihe Model, the bottom width is estimated

316 based on the satellite remote sensing imageries. For the side slope, it is a low sensitive





data, so it could be estimated based on local experiences. For the bottom slope, it is
calculated with the DEM alone the virtual channel section. As there are too many data
for the virtual cross section sizes, so it is not listed in this paper.

**3.3 Parameter optimization**

In Liuxihe Model, an initial parameter set will be derived first based on the terrain
properties, including the DEM, soil type and land use/cover type, then the parameters
will be optimized. In this study, for the insensitive parameter of the land use/cover
related parameters, which is the evaporation coefficient, the initial value is set to be
0.7 for all cells based on the experiences. The initial value of roughness, i.e., the
Manning's coefficient, which is the sensitive parameter of the land use/cover related
parameters, is derived from the land use/cover type based on references (Chen et.al.,
1995, Zhang et.al., 2006, 2007, Shen et.al., 2007, Guo et.al., 2010, Li et.al., 2013,
Zhang et.al., 2015), and listed in Table 2.

Table 2 The initial values of land use/cover related parameters

For the soil related parameters, including the water content at saturation condition, the
water content at field condition, the water content at wilting condition, hydraulic
conductivity at saturation condition, soil thickness and soil porosity characteristics
coefficient b. Based on past modeling experiences and references (Zaradny, 1993,
Anderson et al., 1996), a value of 2.5 is set to b for all soil type, and the water content
at wilting condition is set to be 30% of the water content at saturation condition. The
soil thickness is estimated based on local experiences and listed in Table 3 for all soil
types. The initial values of the water content at saturation condition, the water content
at field condition and hydraulic conductivity at saturation condition are estimated by
using the Soil Water Characteristics Hydraulic Properties Calculator (Arya et al., 1981)
based on soil texture, organic matter, gravel content, salinity and compaction. The
estimated initial values of soil-related parameters are listed in Table 3.

Table 3 The initial values of soil related parameters

In Liuxihe Model, Particle Swarm Optimization(PSO) algorithm (Chen et. al., 2016)





and SCE-UA algorithm (Xu et. al., 2012) were employed to optimize the initial model
parameters. In this study, PSO algorithm is employed to optimize the initial model
parameters as PSO algorithm has been integrated into the Liuxihe Model Cloud (Chen
et. al., 2013). The number of particles of PSO algorithm is set to 20, while the value
range of inertia weight ω is set to 0.1 to 0.9, the value range of acceleration
coefficients C1 is set to 1.25 to 2.75, and C2 to 0.5 to 2.5, and the maximum iteration
is set to 50. Flood event of 20080609 is selected to optimize the parameters of Liuxihe
model, and Fig. 4 shows the result of the parameter optimization. Among them, Fig.
4(a) is the parameters evolving process, Fig. 4(b) is the changing curve of objective
function which is set to minimize the peak flow error, Fig.4(c) is the simulated
hydrograph of flood event 20080609 with the optimized parameters.
Fig. 4 Parameter optimization results of Liuxihe Model for LRB with PSO algorithm
From the results in Fig. 4, it could be found that after 14 evolutions, the parameters
optimization process converges to its optimal values, and the optimal parameters are
achieved, the simulated hydrological process of flood event that is used for parameter
optimization is quite good fitting the observed hydrological process, it could be said
that the parameter has a good optimization effect.

As mentioned above, the automatic parameter optimization of the distributed
hydrological model is very time consuming. In this study, even supercomputer is
employed with parallel computation techniques, the time used for this parameter
optimization is overwhelming, the total time used for achieving the above optimal
parameters of Liuxihe model for LRB flood forecasting is 220 hours, more than 9
days. Considering several runs are usually needed before achieving the final results,
so the parameter optimization procedure may take a few months, this run time is
really a good investment, but the validation results proves this is worth.
**3.4 Model validation**
The other 29 flood events were simulated by using the Liuxihe model with the above





optimized parameters, and the simulated hydrographs of 8 flood events are shown in
Fig. 5, the simulated hydrographs of 8 flood events with initial parameters are also
shown in Fig. 5.

Fig. 5 Simulated flood events by Liuxihe Model with optimized parameters

From the result of Fig. 5, it has been found that the simulated flood processes fits the
observation reasonable well, particularly the simulated peak flow is quite good, and
the simulated hydrological processes with optimized model parameter improved the
simulated hydrological processes largely. To further analyze the effect of parameter
optimization on model performance improvement, five evaluation indices of the
simulated flood events, including the Nash–Sutcliffe coefficient, the correlation
coefficient, the process relative error, the peak flow error and water balance
coefficient are calculated from the simulated results. Table 4 listed the 5 indices for
both the simulated results with the initial parameters and the optimized parameters.

Table 4 Evaluation indices of the simulated flood events

From Table 4, it could be seen that the five evaluation indices are quite good for the
simulated hydrological processes with the optimized model parameters. The average
peak flow error is 5% with 14% the maximum. The average Nash–Sutcliffe
coefficient, correlation coefficient, process relative error and water balance coefficient
are 0.82, 0.83, 0.22 and 0.87 respectively, that are also quite good for large river basin
flood simulation. Five evaluation indices of the simulated hydrological processes with
the optimized model parameters are also good improvements to those simulated with
the initial parameters, those are 0.64, 0.62, 0.37, 0.29 and 0.78. There are excellent
improving in all five indices, with the average increases of 0.18, 0.21 and 0.09 of the
average Nash–Sutcliffe coefficient, correlation coefficient and water balance
coefficient respectively, and the average decreases of the peak flow error and process
relative error are 24% and 15% respectively. So it could be concluded that the Liuxihe
Model set up in LRB with optimized parameters are reasonable and could be used for
flood forecasting of LRB. This also implies that parameter optimization of distributed





hydrological model could improve model performances, and it should be done when it
is possible.

## 5 Results and discussions

### 5.1 Computation time vs model resolution

To evaluate the spatial resolution scaling effect of distributed hydrological modeling
in LRB, the DEM with 90m*90m resolution is rescaled to the resolutions of
500m*500m and 1000m*1000m respectively, the land use and soil type at
1000m*1000m resolution are also rescaled to the same resolutions of the DEM used.
Liuxihe models for LRB flood forecasting at 500m*500m and 1000m*1000m
resolution are set up with the above methods, and the model structures are shown in
Fig. 6.

Fig. 6 Liuxihe Model structure set up for LRB with different resolution

With different spatial resolution, the numbers of grid cells, hill slope cells and river
cells are different, but the river channel order are all set to 3, the numbers of virtual
channel nodes for 500m*500m and 1000m*1000m resolution models are 68 and 33
respectively, numbers of grid cells, hill slope cells and river cells with different model
resolution are listed in Table 5. , the sizes of every virtual cross sections are measured
with the above method.

Table 5 Grid cell numbers with different model spatial resolution

From Table 7, it could be seen, number of grid cells of the model with 200m*200m
resolution is 6.25 times of that with 500m*500m resolution, and 25 times of that with
1000m*1000m resolution, it increases at an approximate exponential of power 2, not
linearly with the model resolution.

Parameters of the models with 500m*500m and 1000m*1000m resolution are
optimized with PSO algorithm by using the same flood event data, and listed in Table
6. From the results it could be seen that some parameters are significantly different



with resolution variation, but some changes little, this implies that the model
parameters are resolution-dependent.
Table 6 Optimized parameters with different model spatial resolution
Computation times required for parameter optimization are quite different. For the
model with 200m*200m resolution, the time for parameter optimization is 220 hours,
while that for models with 500m*500m and 1000m*1000m resolution are 55 and 12
hours respectively. The times needed for parameter optimization of the model at
200m*200m resolution is 4 times of that for 500m*500m resolution model and 18.3
times of that for 1000m*1000m resolution model respectively. Considering the time
needed for model run, the 200m*200m model resolution is regarded as appropriate for
LRB.
**5.2 Model performance vs model resolution**
The other 29 flood events are also simulated with the models at 500m*500m
resolution and 1000m*1000m resolution. Simulated hydrograph of 5 flood events,
including 2 big, 2 medium and one small ones are shown in Fig. 7.
Fig. 7 Simulated results with different model resolutions
From the results it could be seen that the simulated hydrological processes with 3
different spatial resolutions are quite different. The result simulated with
1000m*1000m resolution is not so good, although the flood shapes are simulated well,
but the peak flow are much lower than that of the observation, so the result is not
acceptable, and could not be recommended. The result simulated with 500m*500m
resolution model is a big improvement to that simulated with 1000m*1000m
resolution model, the flood shapes are more similar to the observation, and the peak
flow is also get closer to the observation, and could be recommended for flood
forecasting if the spatial resolution could not be much finer. The result simulated with
200m*200m resolution model is a further improvement to that simulated with
500m*500m resolution model, the flood shapes fits the observation much better, and
the peak flows are also much closer to the observation also, it is the good simulation





results and could be recommended for flood forecasting of LRB. The results are good
enough that there is no need to further explore the finer model resolution.

## 6 Conclusions

By employing Liuxihe Model, a physically based distributed hydrological model, this
study sets up a distributed hydrological model for the flood forecasting of Liujiang
River Basin in southern China that could be regarded as a large watershed. Terrain
data including DEM, soil type and land use type are downloaded from the website
freely, and the model structure with a high resolution of 200m*200m grid cell is set
up, which divides the whole watershed into 1469900 grid cells that is further divided
into 1463204 hill slope cells and 6696 river cells. The initial model parameters are
derived from the terrain property data, and then optimized by using the PSO algorithm
with one observed flood event, which improves the model performance largely. 29
observed flood events are simulated by using the model with optimized parameters,
the results are analyzed, and the model scaling effects are studied. Based on these
studies, following conclusions are suggested.

1. In Liuxihe Model, the river channels are divided into virtual channel sections, and
the cross section shapes are assumed to be trapezoid and the size is the same within
the virtual channel section. The size of the virtual channel section is simplified to
three indices, including bottom width, side slope and bottom slope, those are
estimated by using remote sensing imageries. This method not only makes the
distributed model application practical, but also simplifies the river channel routing
method. This significantly increases the model computation efficiency, and makes it
could be used in larger watersheds. Results in this study shows the model setting up
with this method has a reasonable performance, i.e., this simplification has not
sacrificed the model's flood simulation accuracy significantly, so this simplification
could be used in large watershed distributed hydrological modelling, including
Liuxihe model and other models.



2. Uncertainty exists for physically derived model parameters. Parameter optimization
could reduce parameter uncertainty, and is highly recommended to do so when there
is some observed hydrological data. In this study, the simulated hydrograph with
optimized model parameters is more fitting the observed hydrograph in shape than
that simulated with initial model parameters, the 5 evaluation indices are improved
also. The average increases of Nash–Sutcliffe coefficient, correlation coefficient and
water balance coefficient are 0.18, 0.21 and 0.09 respectively, the average decreases
of the peak flow error and process relative error are 24% and 15% respectively, this
implies that the model performance is improved significantly with parameter
optimization.

3. Computation time needed for running a distributed hydrological model increases
exponentially at an approximate power of 2, not linearly with the increasing of model
spatial resolution. In this study, the computation time required for parameter
optimization for the model with 200m*200m resolution is 220 hours, that is 4 times of
that of the model at 500m*500m and 18.3 times of that of the model at 1000m*1000m
resolution respectively. Based on the Liuxihe Model cloud system implemented on the
high performance supercomputer, the 200m*200m model resolution is the highest
resolution that could be fulfilled in modeling Liujiang River Basin flooding with
Liuxihe Model considering the computation cost. This also means that if the user
could pay high computation cost, then larger watershed could also be modelled with
Liuxihe Model by implemented the Liuxihe Model cloud system on a much more
advanced high performance supercomputer, this could be easily done nowadays if the
user thinks this investment is a worth doing.

4. In forecasting watershed flood by using distributed hydrological model, minimum
model spatial resolution needs to be maintained to keeping the model an acceptable
performance. Usually if the model spatial resolution increases, i.e., the grid cell gets
smaller, the model performance is better, but this will increase the run time



significantly, so there is a threshold model spatial resolution to keep the model
performance reasonable while keep the model run at the least time. In this study, the
threshold model spatial resolution is at 500m*500m grid cell, but the resolution at
200m*200m grid cell is recommended by trading-off between the computation cost
and the model performance. This conclusion may be different in different watersheds
for Liuxihe Model, or even different in the same watershed for different models.

5. Terrain data downloaded freely from the website derived the river channel system
that is very similar to the natural river channel system after it is rescaled from its
original spatial resolution of 90m*90m to 200m*200m, 500m*500m and
1000m*1000m, but the higher resolution DEM describes the river channel more in
details. This means that the freely downloaded DEM could be used to set up the
Liuxihe Model for Liujiang River Basin flood forecasting.

***Acknowledgements*:** This study is supported by the Special Research Grant for the
Water Resources Industry (funding no. 201301070), the National Science Foundation
of China (funding no. 50479033), and the Basic Research Grant for Universities of
the Ministry of Education of China (funding no. 13lgjc01).





**Figures**

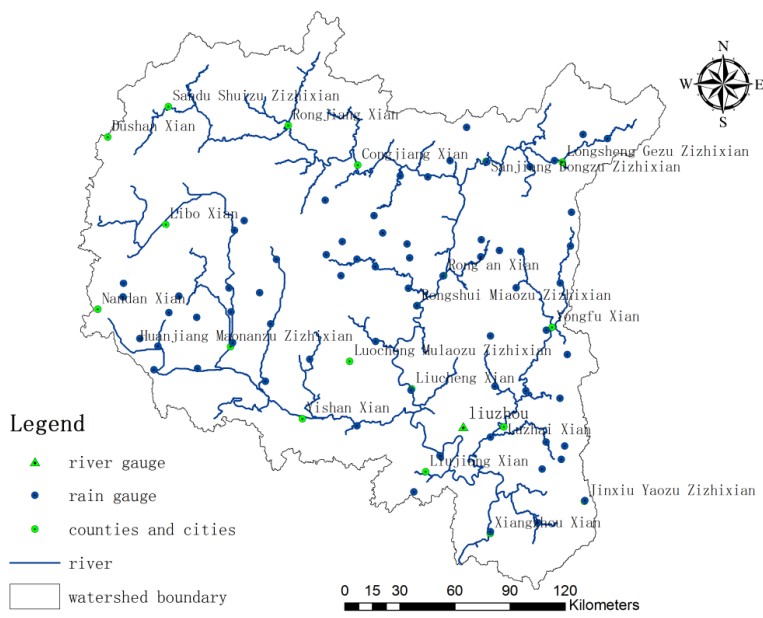


Fig. 1 sketch map of Liujiang River Basin

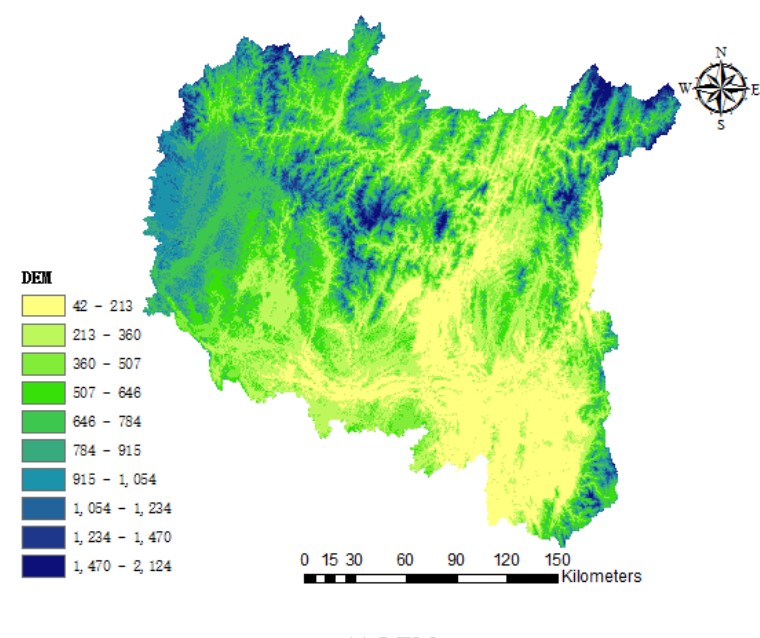


(a) DEM



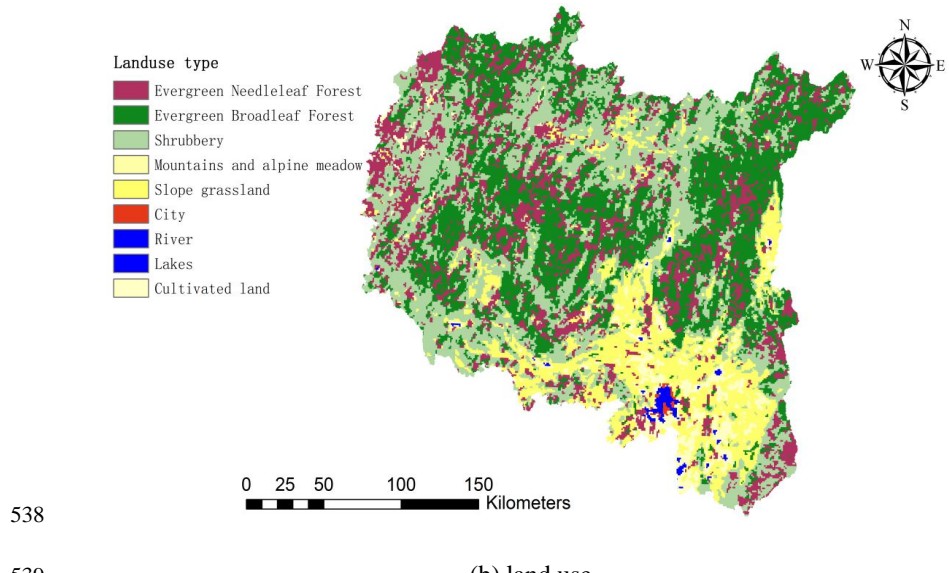


(b) land use

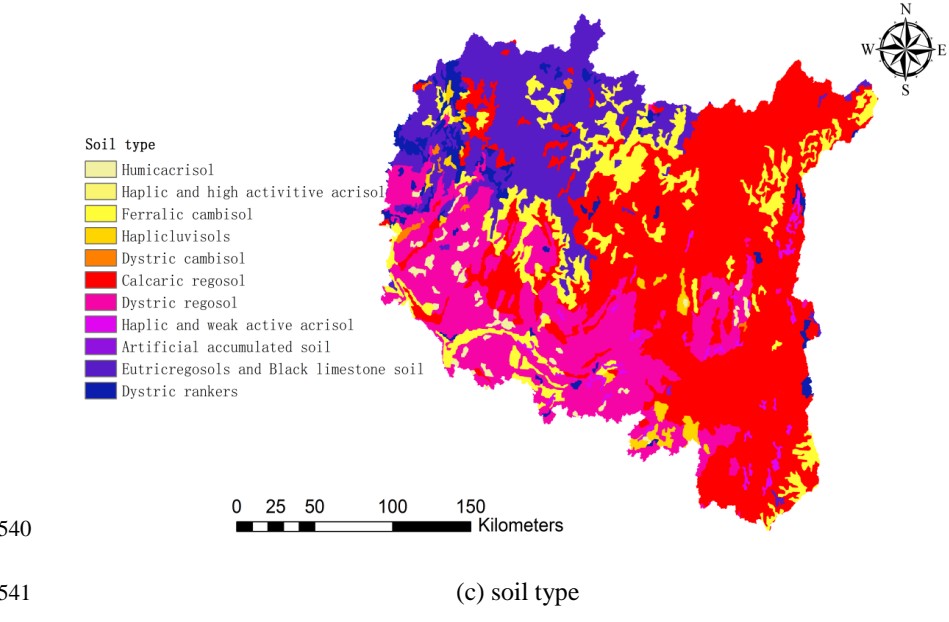


(c) soil type

Fig. 2 Terrain properties of LRB






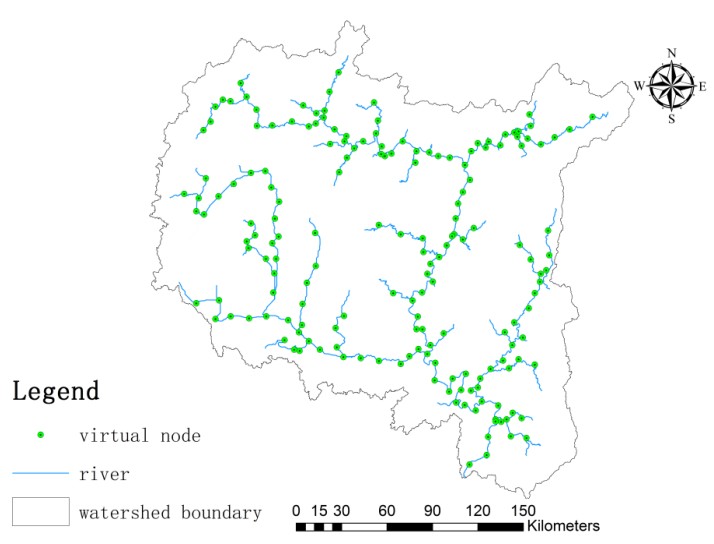


Fig. 3 Liuxihe Model structure set up for LRB (200m×200m resolution)


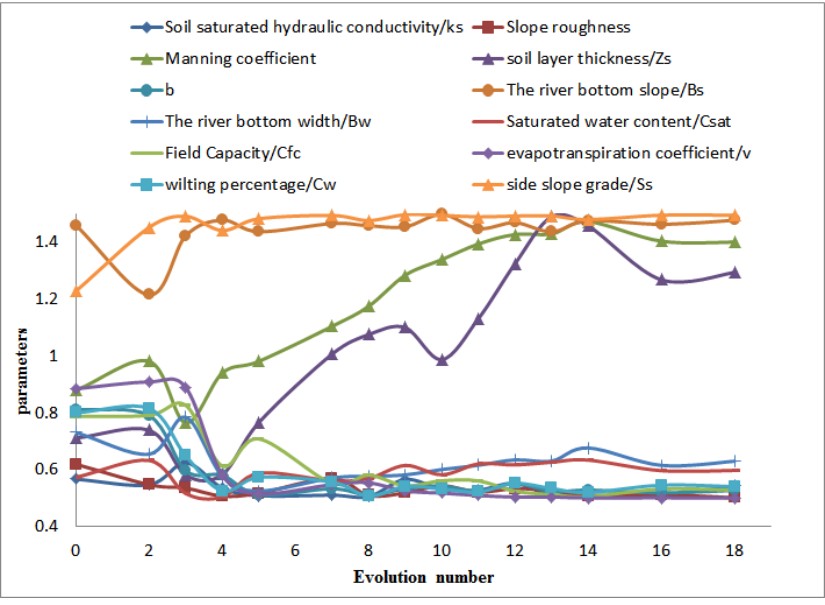

(a) Parameter evolution process



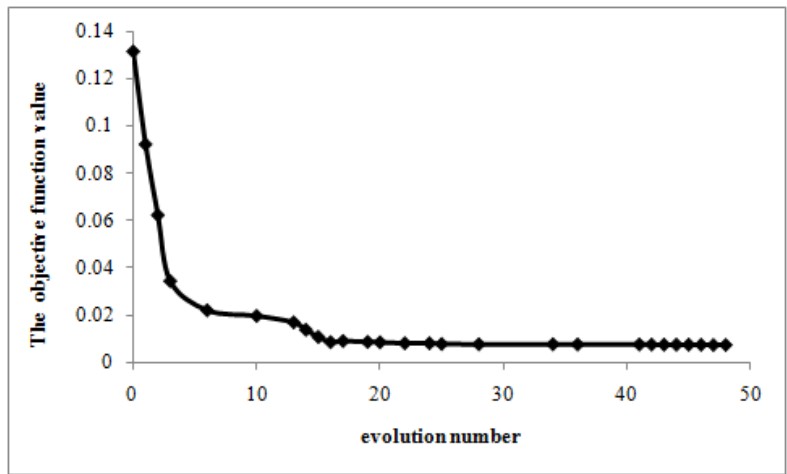

(b) Changing curve of objective function

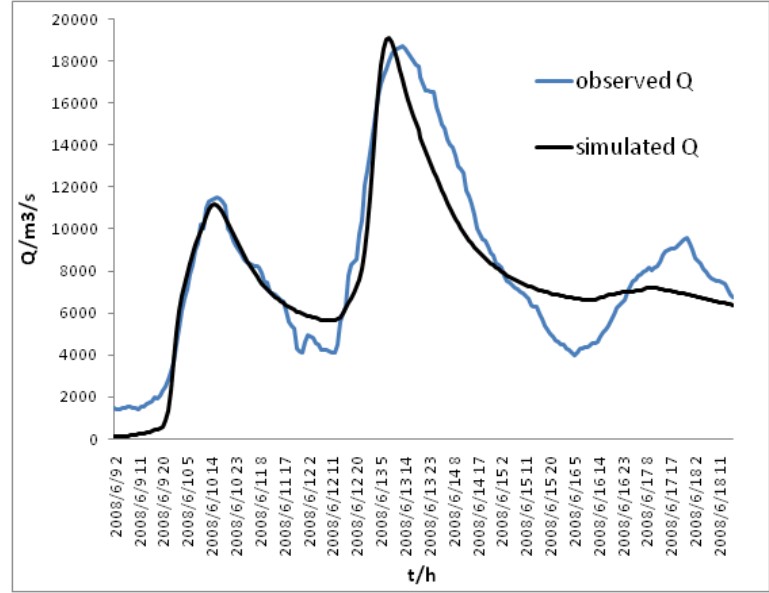

(c) Simulated flood process
Fig. 4 Parameter optimization results of Liuxihe Model for LRB with PSO algorithm




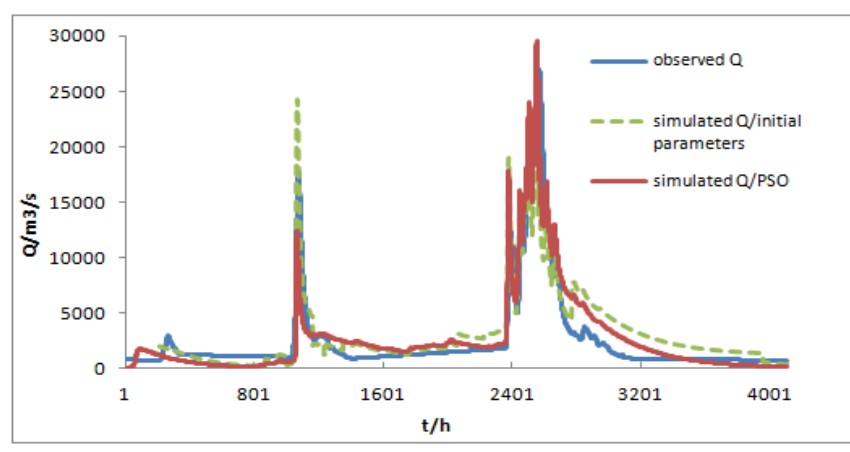

(a) flood event 1988051620

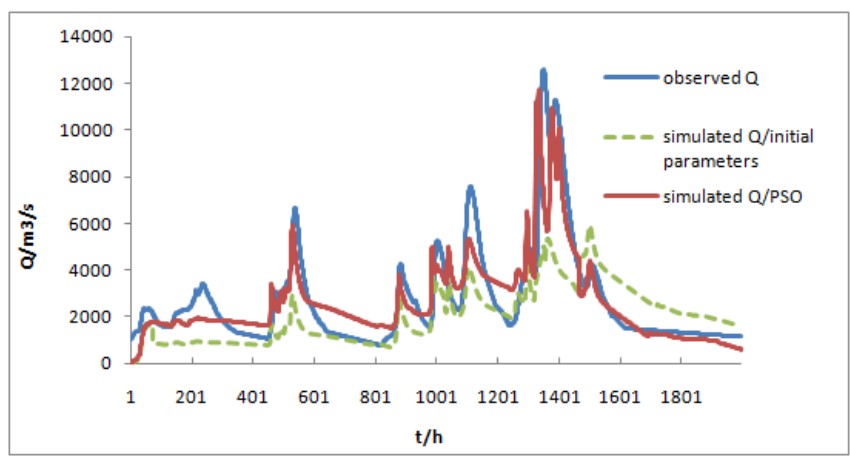

(b) flood event 1982042116

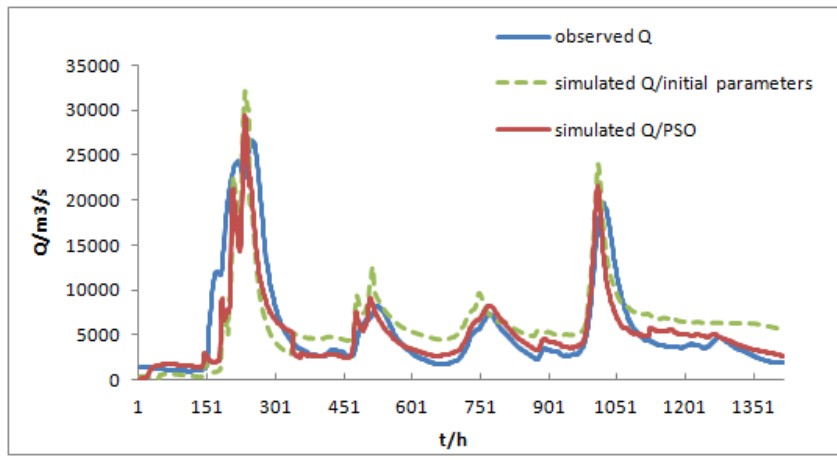

(c) flood event 1994060700






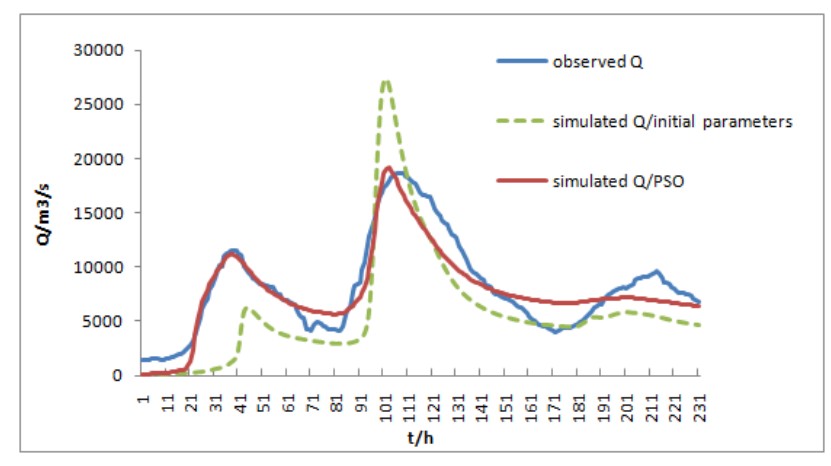

(d) flood event 2008060902


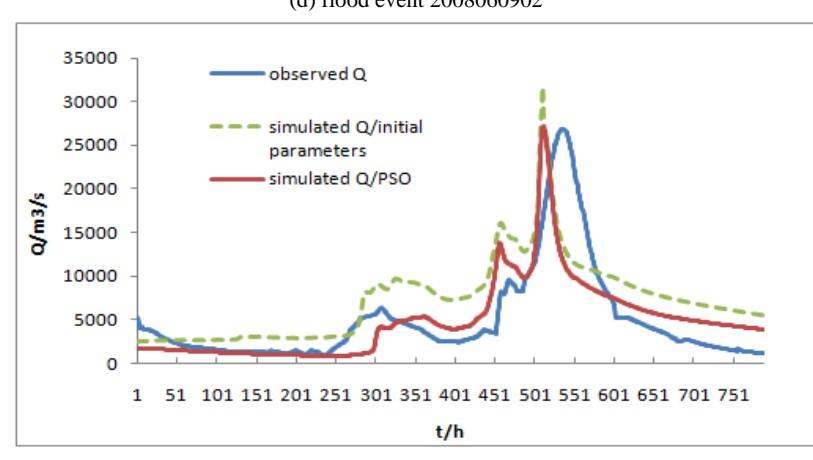

(e) flood event 200906090800


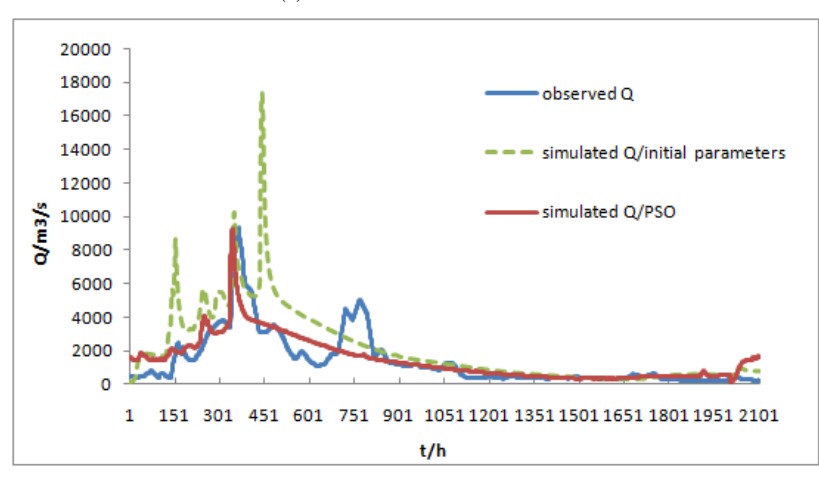

(f) flood event 201106010900





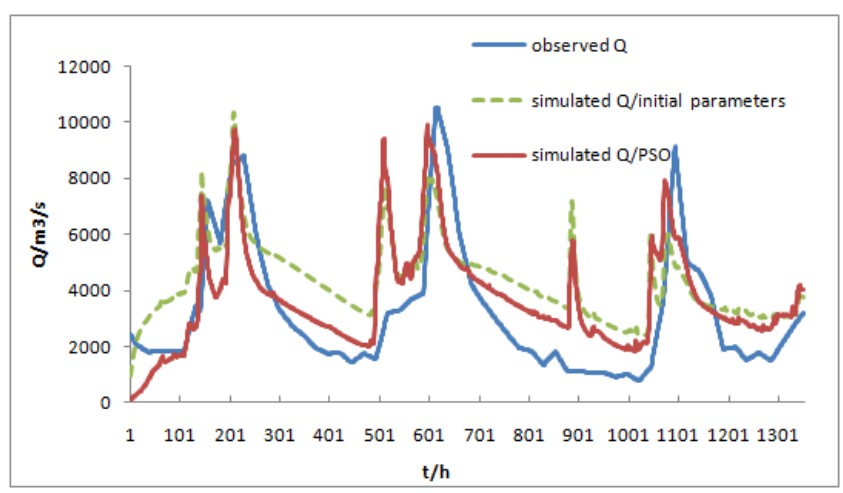


(g) flood event 201206022000

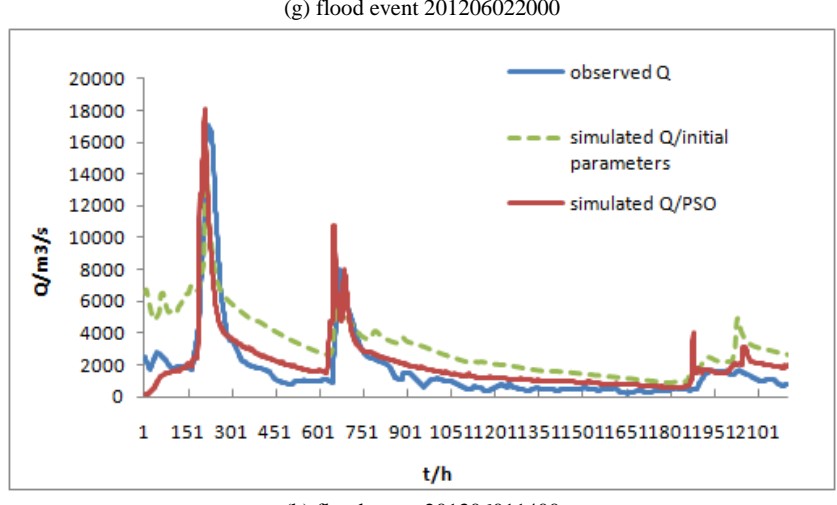

(h) flood event 201306011400
Fig. 5 Simulated flood events by Liuxihe Model with optimized parameters











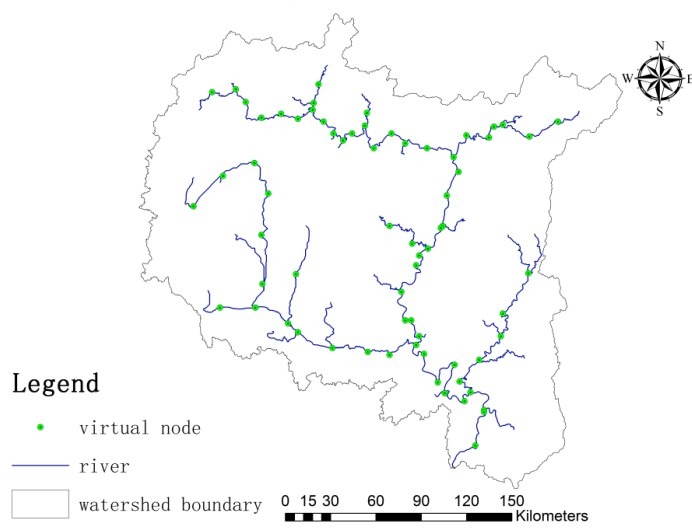


(a) 500m×500m resolution

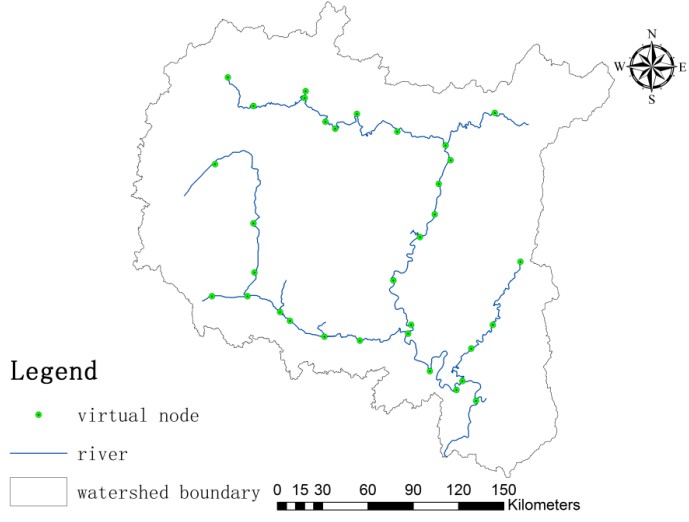


(b) 1000m×1000m resolution

Fig. 6 Liuxihe Model structure set up for LRB with different resolution






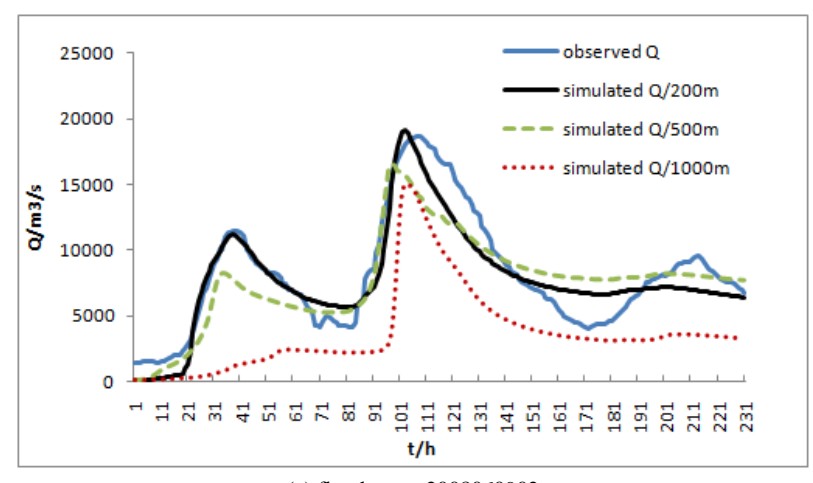

(a) flood event 2008060902

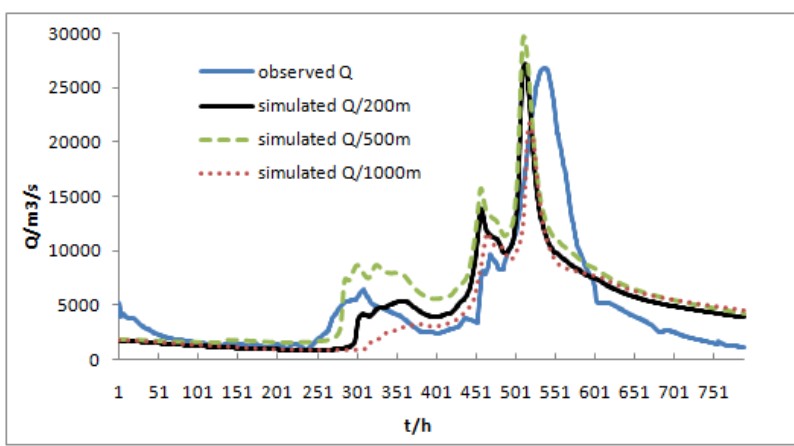


(b) flood event 2009060908

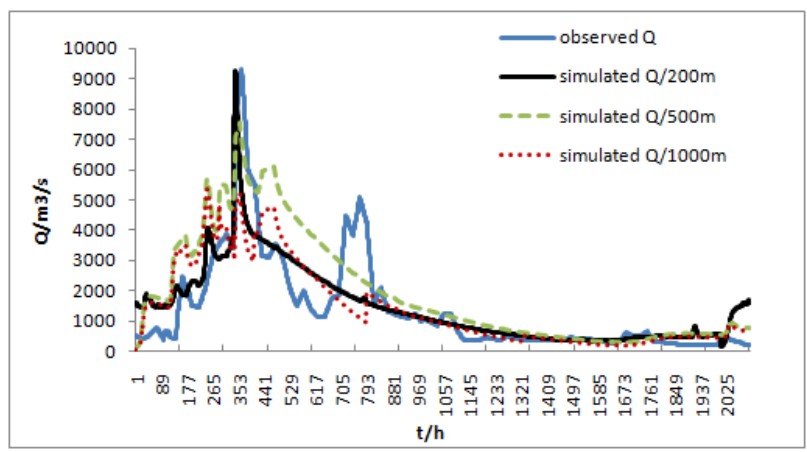

(c) flood event 2011060109





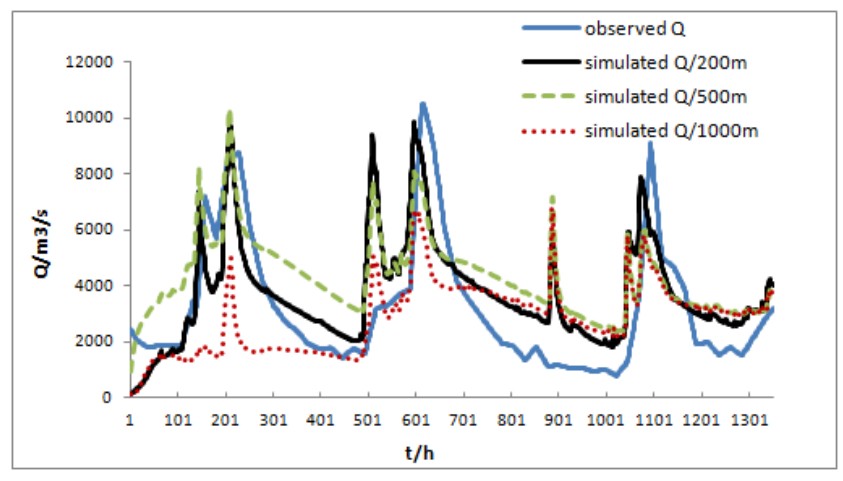

(d) flood event 2012060220

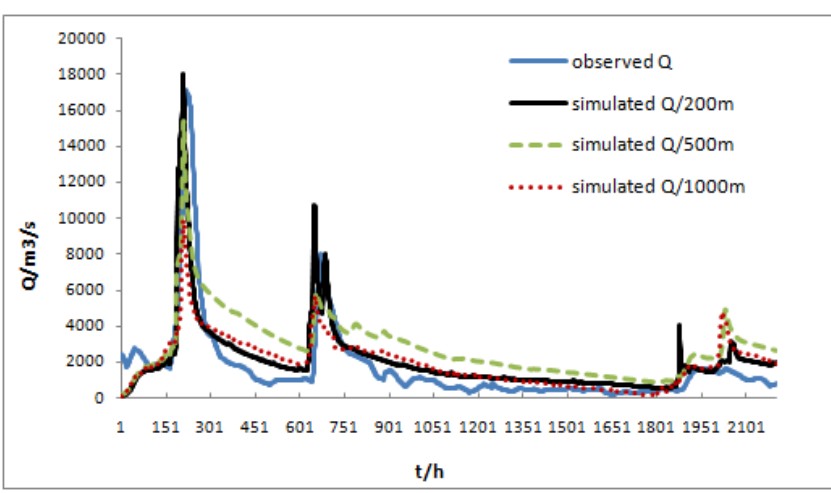

(e) flood event 2013060114

Fig. 7 Simulated results with different model resolutions






**Tables**

Table1 Brief information of flood events in LRB

| No. | Floods No. | Start time (yyyymmddhh) | End time (yyyymmddhh) | length of time/h | peak flow (m³/s) |
|---|---|---|---|---|---|
| 1 | 1982042116 | 1982042116 | 1982110216 | 4614 | 12600 |
| 2 | 1983020308 | 1983020308 | 1983021722 | 350 | 7880 |
| 3 | 1984021100 | 198402100 | 1984040105 | 1205 | 12900 |
| 4 | 1985011900 | 1985011900 | 1985021114 | 544 | 11400 |
| 5 | 1986022300 | 1986022300 | 1986042004 | 1334 | 12200 |
| 6 | 1987050100 | 1987050100 | 1987071700 | 1848 | 10800 |
| 7 | 1988070620 | 1988070620 | 1988100605 | 2915 | 27000 |
| 8 | 1989042600 | 1989042600 | 1989081009 | 2499 | 7500 |
| 9 | 1990050100 | 1990001000 | 1990072306 | 2006 | 11400 |
| 10 | 1991053118 | 1991053118 | 1991062806 | 686 | 14300 |
| 11 | 1992042900 | 1992042900 | 1992072107 | 1977 | 18100 |
| 12 | 1993060900 | 1993060900 | 1993082408 | 1818 | 21200 |
| 13 | 1994060700 | 1994060700 | 1994080706 | 1416 | 26500 |
| 14 | 1995052100 | 1995052100 | 1995071506 | 1296 | 17300 |
| 15 | 1996060600 | 1996060600 | 1996081808 | 1728 | 33700 |
| 16 | 1997060400 | 1997060400 | 1997062406 | 476 | 13600 |
| 17 | 1998051600 | 1998051600 | 1998090100 | 2520 | 19600 |
| 18 | 1990050100 | 1999050100 | 1999080404 | 1134 | 17800 |
| 19 | 2000052100 | 2000052100 | 2000061809 | 659 | 24100 |
| 20 | 2001051500 | 2001051500 | 2001062300 | 910 | 14200 |
| 21 | 2002042600 | 2002042600 | 2002081000 | 2520 | 17900 |
| 22 | 2003060600 | 2003060600 | 2003072103 | 843 | 11600 |
| 23 | 2004070300 | 200407000 | 2004081508 | 998 | 23700 |
| 24 | 2005061400 | 2005061400 | 2005070702 | 552 | 16400 |
| 25 | 2006060400 | 2006060400 | 2006071000 | 870 | 13200 |
| 26 | 2008060900 | 2008060900 | 2008061908 | 238 | 18700 |
| 27 | 2009060908 | 2009060908 | 2009071208 | 788 | 26800 |
| 28 | 2011061090 | 2011061009 | 2011090104 | 2004 | 9153 |
| 29 | 2012060220 | 2012060220 | 2012080101 | 1351 | 10500 |
| 30 | 2013060114 | 2013060114 | 2013090114 | 2200 | 17100 |










Table 2 The initial values of land use/cover related parameters

| Land use/cover | evaporation coefficient | roughness coefficient |
|---|---|---|
| Evergreen needle leaf forest | 0.7 | 0.4 |
| Evergreen broadleaf forest | 0.7 | 0.6 |
| Shrubbery | 0.7 | 0.4 |
| Mountains and alpine meadow | 0.7 | 0.2 |
| Slope grassland | 0.7 | 0.3 |
| City | 0.7 | 0.05 |
| Cultivated land | 0.7 | 0.35 |


Table 3 The initial values of soil related parameters

| Soil Type | soil thickness | water content at saturation condition | water content at field condition | hydraulic conductivity at saturation condition |
|---|---|---|---|---|
| Humicacrisol | 800 | 0.65 | 0.32 | 3.5 |
| Haplic and high active acrisol | 900 | 0.57 | 0.43 | 4.2 |
| Ferralic cambisol | 850 | 0.63 | 0.38 | 20.5 |
| Haplicluvisols | 980 | 0.46 | 0.15 | 2.6 |
| Dystric cambisol | 950 | 0.55 | 0.41 | 14 |
| Calcaric regosol | 1100 | 0.62 | 0.24 | 5.6 |
| Dystric regosol | 840 | 0.45 | 0.27 | 12.5 |
| Haplic and weak active acrisol | 1050 | 0.58 | 0.16 | 4.6 |
| Artificial accumulated soil | 1000 | 0.63 | 0.34 | 5.5 |
| Eutricregosols and Black limestone | 550 | 0.75 | 0.27 | 3.5 |
| Dystric rankers | 380 | 0.78 | 0.36 | 8 |











629          Table 4 Evaluation indices of the simulated flood events

| ID | floods | parameters | Nash–Sutcliffe coefficient/C | Correlation coefficient/R | Process relative error/P | Peak flow relative error/E | Water balance coefficient/W |
|----|--------|------------|------------------------------|---------------------------|--------------------------|----------------------------|-----------------------------|
| 1 | 1982081219 | initial | 0.52 | 0.48 | 0.56 | 0.58 | 0.52 |
| | | optimized | 0.84 | 0.75 | 0.30 | 0.01 | 0.83 |
| 2 | 1983020308 | initial | 0.60 | 0.55 | 0.45 | 0.26 | 0.65 |
| | | optimized | 0.82 | 0.84 | 0.21 | 0.04 | 0.89 |
| 3 | 1984010100 | initial | 0.62 | 0.71 | 0.38 | 0.32 | 0.75 |
| | | optimized | 0.75 | 0.89 | 0.26 | 0.14 | 0.96 |
| 4 | 1985010100 | initial | 0.58 | 0.57 | 0.35 | 0.33 | 0.85 |
| | | optimized | 0.73 | 0.87 | 0.17 | 0.01 | 1.05 |
| 5 | 1986010100 | initial | 0.65 | 0.62 | 0.38 | 0.25 | 0.62 |
| | | optimized | 0.83 | 0.85 | 0.23 | 0.04 | 0.94 |
| 6 | 1987050100 | initial | 0.76 | 0.45 | 0.35 | 0.36 | 0.58 |
| | | optimized | 0.93 | 0.76 | 0.10 | 0.05 | 1.01 |
| 7 | 19880516200 | initial | 0.54 | 0.58 | 0.26 | 0.42 | 0.82 |
| | | optimized | 0.84 | 0.80 | 0.15 | 0.04 | 0.90 |
| 8 | 1989042600 | initial | 0.52 | 0.55 | 0.55 | 0.25 | 0.62 |
| | | optimized | 0.64 | 0.74 | 0.39 | 0.02 | 0.88 |
| 9 | 1990050100 | initial | 0.55 | 0.64 | 0.42 | 0.23 | 0.55 |
| | | optimized | 0.85 | 0.87 | 0.14 | 0.03 | 0.85 |
| 10 | 1991053118 | initial | 0.63 | 0.62 | 0.40 | 0.18 | 0.68 |
| | | optimized | 0.80 | 0.76 | 0.25 | 0.04 | 0.95 |
| 11 | 1992042900 | initial | 0.48 | 0.59 | 0.35 | 0.34 | 0.65 |
| | | optimized | 0.66 | 0.84 | 0.20 | 0.11 | 0.89 |
| 12 | 1993060900 | initial | 0.75 | 0.65 | 0.38 | 0.28 | 0.84 |
| | | optimized | 0.91 | 0.89 | 0.24 | 0.09 | 1.05 |
| 13 | 1994060700 | initial | 0.78 | 0.64 | 0.32 | 0.26 | 1.25 |
| | | optimized | 0.93 | 0.85 | 0.14 | 0.04 | 0.85 |
| 14 | 1995052100 | initial | 0.68 | 0.48 | 0.42 | 0.35 | 0.65 |
| | | optimized | 0.82 | 0.70 | 0.20 | 0.01 | 0.81 |
| 15 | 1996060600 | initial | 0.74 | 0.65 | 0.25 | 0.23 | 0.54 |
| | | optimized | 0.90 | 0.93 | 0.18 | 0.02 | 0.86 |
| 16 | 1997060400 | initial | 0.65 | 0.51 | 0.23 | 0.26 | 0.65 |
| | | optimized | 0.84 | 0.87 | 0.13 | 0.06 | 0.95 |
| 17 | 1998051600 | initial | 0.57 | 0.62 | 0.35 | 0.18 | 0.68 |
| | | optimized | 0.83 | 0.85 | 0.30 | 0.01 | 1.05 |
| 18 | 1999061700 | initial | 0.48 | 0.59 | 0.33 | 0.15 | 0.55 |
| | | optimized | 0.60 | 0.83 | 0.15 | 0.05 | 0.80 |
| 19 | 2000052100 | initial | 0.67 | 0.62 | 0.45 | 0.25 | 0.58 |
| | | optimized | 0.79 | 0.89 | 0.26 | 0.06 | 0.83 |
| 20 | 2001051500 | initial | 0.62 | 0.56 | 0.32 | 0.22 | 0.68 |





| | | | | | | | |
|---|---|---|---|---|---|---|---|
| | | optimized | 0.80 | 0.82 | 0.25 | 0.07 | 0.82 |
| 21 | 2002042600 | initial | 0.68 | 0.65 | 0.38 | 0.18 | 0.57 |
| | | optimized | 0.86 | 0.90 | 0.24 | 0.02 | 0.87 |
| 22 | 2003060600 | initial | 0.75 | 0.55 | 0.25 | 0.26 | 0.55 |
| | | optimized | 0.92 | 0.85 | 0.14 | 0.04 | 0.76 |
| 23 | 2004070300 | initial | 0.58 | 0.68 | 0.38 | 0.27 | 0.68 |
| | | optimized | 0.78 | 0.82 | 0.23 | 0.08 | 0.85 |
| 24 | 2005061400 | initial | 0.65 | 0.62 | 0.52 | 0.32 | 0.65 |
| | | optimized | 0.76 | 0.76 | 0.35 | 0.06 | 0.74 |
| 25 | 2006060400 | initial | 0.68 | 0.72 | 0.62 | 0.35 | 0.53 |
| | | optimized | 0.82 | 0.83 | 0.30 | 0.10 | 0.86 |
| 26 | 2009060908 | initial | 0.75 | 0.78 | 0.25 | 0.23 | 1.22 |
| | | optimized | 0.95 | 0.92 | 0.17 | 0.04 | 0.09 |
| 27 | 2011010100 | initial | 0.66 | 0.75 | 0.35 | 0.55 | 1.66 |
| | | optimized | 0.80 | 0.84 | 0.26 | 0.03 | 1.02 |
| 28 | 2012010100 | initial | 0.63 | 0.68 | 0.34 | 0.22 | 1.42 |
| | | optimized | 0.82 | 0.79 | 0.20 | 0.05 | 0.80 |
| 29 | 2013010100 | initial | 0.78 | 0.65 | 0.31 | 0.32 | 1.35 |
| | | optimized | 0.95 | 0.82 | 0.20 | 0.06 | 0.92 |
| | average | initial | 0.64 | 0.62 | 0.37 | 0.29 | 0.78 |
| | | optimized | 0.82 | 0.83 | 0.22 | 0.05 | 0.87 |




Table 5 Grid cell numbers with different model spatial resolution

| model resolution | Number of grid cells | Number of hill slope cells | Number of river cells |
|---|---|---|---|
| 200m*200m | 1469900 | 1463204 | 6696 |
| 500m*500m | 235184 | 234113 | 1071 |
| 1000m*1000m | 58796 | 58528 | 268 |









Table 6 Optimized parameters with different model spatial resolution

| Resolution | Soil saturated hydraulic conductivity/ks | Slope roughness | Manning coefficient | Soil layer thickness/Zs | b | The river bottom slope/Bs |
|---|---|---|---|---|---|---|
| | 1.33 | 0.66 | 1.19 | 1.42 | 0.67 | 0.75 |
| 200m | The river bottom width/Bw | Saturated water content/C sat | Field Capacity/ Cfc | Evapotranspiration coefficient/v | Wilting percentage/ Cw | Side slope grade/Ss |
| | 1.24 | 1.11 | 1.2 | 0.94 | 0.68 | 1.42 |
| | Soil saturated hydraulic conductivity/ks | Slope roughness | Manning coefficient | Soil layer thickness/Zs | b | The river bottom slope/Bs |
| 500m | 0.67 | 1.47 | 1.49 | 1.37 | 1.5 | 0.51 |
| | The river bottom width/Bw | Saturated water content/C sat | Field Capacity/ Cfc | Evapotranspiration coefficient/v | Wilting percentage/ Cw | Side slope grade/Ss |
| | 0.91 | 1.16 | 1.41 | 1.37 | 1.37 | 0.5 |
| | Soil saturated hydraulic conductivity/ks | Slope roughness | Manning coefficient | Soil layer thickness/Zs | b | The river bottom slope/Bs |
| 1000m | 0.5 | 1.43 | 1.17 | 1.11 | 1.47 | 0.57 |
| | The river bottom width/Bw | Saturated water content/C sat | Field Capacity/ Cfc | Evapotranspiration coefficient/v | Wilting percentage/ Cw | Side slope grade/Ss |
| | 1.1 | 0.76 | 0.53 | 0.6 | 1.5 | 0.54 |







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
