# Peer review of "Large watershed flood forecasting with high resolution distributed hydrological model"

_Hydrology and Earth System Sciences, 2016_

## Referee Comment (RC1) · Anonymous Referee #1 · 29 Sep 2016

In this work, a physically based distributed hydrological model was used for flood fore-casting in a large watershed to validate the feasibility of distributed hydrological model's application for large watershed flood forecasting. The research objective is significant. A suitable revision is needed before it can be accepted for publication by HESS, and the following comments below should be addressed: 1. It is insufficient that discharge from only one station were used for validation in such a large watershed with 58270 km2. Because the same effect may come from different combinations of parameters, more hydrological variables need to be checked seriously. Hence, it is suggested that validation with more observation at different river locations in the area should be added or that spatial observations, such as evaporation and soil moisture from satellite data, could be utilized for check the model performance in this large watershed. 2. In sec-tion "Parameter optimization", some are unclear. Are the parameters of PSO fixed for

once? What is the objective function? As the paper tells it is set to minimize the peak flow error, but from the Fig 4(c), the peaks have not yet been captured well enough. The result of optimization could be not the real optimal. More trainings are needed. In addition, Nash–Sutcliffe coefficient may be more suitable for the objective function. 3. How did the simulation consider the reservoir regulation in the work? 4. The authors compared model performance with model resolutions at 200m, 500m, 1000m. Although the result is significant, it is better to add some comparisons with model resolutions with smaller interval, such as 300m or 400m. This manuscript does not explain the reason why this work has just chosen those model resolutions. 5. In this paper, the abstract should be more concise and the motivation is not very clear. 6. From line 116 to 133, the related works should be classified and then summarized concisely. 7. If this work did not modify the Liuxihe Model pubulished in the previous works, I suggest section 3.1 and 3.2 should be merged into one part. The description of the model could be reduced. Some contents in Section "introduction" and 3.1 are repeated. 8. The coordinate information in the maps of Fig. 1-3 and Fig. 6 should be displayed. The plotting scales should be the same for all maps. 9. The font in some figures should be accord with that in the manuscript. The units in some figures look unprofessional. For example, in Fig. 4(c), the title of the x axis should be "date" without unit. 10. The trend lines in Fig. 5 and 7 are not clear, especially during the flood. All the plots should be rearranged in the panels.

---

## Referee Comment (RC2) · Anonymous Referee #2 · 2 Dec 2016

This manuscript discusses an interesting topic of using a physically-based distributed model for flood forecasting in large watershed. It deals with challenges of computational speed and forecasting accuracy, possibly advancing flood mitigation and control decisions in practice. Following suggestions will help the authors improve the manuscript:

(1) Sections 3.2 ∼ 3.4 need to be reorganized by moving the results of the model (e.g., calibration and validation performances) to a subsection of "5. Results." These three sections can be combined into one subsection to just discuss the approaches for model calibration and validation.

(2) A section of Discussion is lacking. This section should be added to thoroughly discuss the major findings of this study and assess these findings in accordance with those reported by others. The purpose of this section is to put this study on the international arena.

(3) Both section Abstract and Conclusions are too long. They need to be rewritten to concisely highlight the importance/scientific contributions of the study, study approaches, and major findings. Some generic statements can be deleted.

(4) Appropriate details of formalization should be provided to help audiences better understand your contributions. For example, the descriptions of the model core components, including saturation excess method, kinematic wave approximation, diffusive wave approximation, and particle swarm optimization (PSO) algorithm, are too simple. Some necessary formulas/equations should be presented because these methods could have distinctly different complexity levels, depending on how they are formulated.

(5) More descriptions should be provided for the statistics (e.g., Nash-Sutclife coefficient and water balance coefficient). What are the thresholds of these statistics, above which the model can be judged to have a good performance?

(6) Some subject terminologies should be clarified. For example, the manuscript uses several different adjectives, such as "disastrous", "serious", "huge", and "large", to describe flood magnitude. How do you classify the floods, in terms life or economic losses?

(7) In Table 6 and other tables/figures, the units for some quantities are missing. Also, the values of Manning's n are larger than one. How can this be possible? In Table 6, you reported Manning's n = 1.17 $\sim$ 1.49.

(8) Lines 292-294: if observed data are not available, how can you optimize the model parameters? What to be optimized?

(9) The annotated manuscript has some specific comments. I suggest that the manuscript be proofread by an English native speaker. There are a number of awkward phrases/words and confusing sentences.

Please also note the supplement to this comment:
http://www.hydrol-earth-syst-sci-discuss.net/hess-2016-489/hess-2016-489-RC2-
supplement.pdf

─────────────────────────

[Figure]

**Supplement:**

[revised manuscript text omitted]

**Figures**

[Figure]

Fig. 1 sketch map of Liujiang River Basin

[Figure]

    (a) DEM

                           (b) land use

[Figure]

                           (c) soil type

                Fig. 2 Terrain properties of LRB

[Figure]

[Figure]

Fig. 3 Liuxihe Model structure set up for LRB (200m×200m resolution)

[Figure]

              (a) Parameter evolution process

          (b) Changing curve of objective function

[Figure]

              (c) Simulated flood process
Fig. 4 Parameter optimization results of Liuxihe Model for LRB with PSO algorithm

[Figure]

[Figure]

[Figure]

                  (a) flood event 1988051620

                  (b) flood event 1982042116

                  (c) flood event 1994060700

                     (d) flood event 2008060902

                     (e) flood event 200906090800

                     (f) flood event 201106010900

(g) flood event 201206022000

[Figure]

(h) flood event 201306011400

Fig. 5 Simulated flood events by Liuxihe Model with optimized parameters

[Figure]

[Figure]

[Figure]

              (a) 500m×500m resolution

[Figure]

              (b) 1000m×1000m resolution

Fig. 6 Liuxihe Model structure set up for LRB with different resolution

[Figure]

[Figure]

[Figure]

  (a) flood event 2008060902

  (b) flood event 2009060908

  (c) flood event 2011060109

(d) flood event 2012060220

[revised manuscript text omitted]

---

## Author Comment (AC1) · 8 Dec 2016

Anonymous Referee #1 In this work, a physically based distributed hydrological model was used for flood forecasting in a large watershed to validate the feasibility of distributed hydrological model's application for large watershed flood forecasting. The research objective is significant. A suitable revision is needed before it can be accepted for publication by HESS, and the following comments below should be addressed:

Reply: Thank the reviewer for his/her comments, revisions have be down based on the reviewer's comments. Following are responses to the reviewer's comments one by one.

1. It is insufficient that discharge from only one station were used for validation in such a large watershed with 58270km2. Because the same effect may come from different

combinations of parameters, more hydrological variables need to be checked seriously. Hence, it is suggested that validation with more observation at different river locations in the area should be added or that spatial observations, such as evaporation and soil moisture from satellite data, could be utilized for check the model performance in this large watershed.

Reply: If there are more information available, such as the reviewer recommended, it will surely improve the model performance, but in practice, the mostly available data is the discharge at the watershed outlet, it is the available data for most large watersheds. Results in this paper has shown with this data, the model parameters could be optimized reasonably well, and the practice in this paper is acceptable. For Liujiang watershed studied in this article, there is no other observation data, so the authors will not be able to do more works on this aspect.

2. In section "Parameter optimization", some are unclear. Are the parameters of PSO fixed for once? What is the objective function? As the paper tells it is set to minimize the peak flow error, but from the Fig 4(c), the peaks have not yet been captured well enough. The result of optimization could be not the real optimal. More trainings are needed. In addition, Nash–Sutcliffe coefficient may be more suitable for the objective function.

Reply: The parameters after optimization will be fixed, but they could be re-optimized if there are new data, or with different optimization strategies. Different objective function could be employed, it is flexible. Actually, we have tested a few objective functions including minimizing Nash–Sutcliffe coefficient, maximizing correlation coefficient, minimizing process relative error, minimizing peak flow relative error and maximizing water balance coefficient. The results shown that with the objective function of minimizing peak flow relative error, the overall accuracy for peak flow is the best one, that fits the large watershed flood forecasting's concern, so the objective function of minimizing peak flow relative error is employed.

[Figure]

3. How did the simulation consider the reservoir regulation in the work?

Reply: In Liujiang watershed, there are no big reservoirs with significant regulation capacity, so in this study, reservoir regulation was not considered.

4. The authors compared model performance with model resolutions at 200m, 500m, 1000m. Although the result is significant, it is better to add some comparisons with model resolutions with smaller interval, such as 300m or 400m. This manuscript does not explain the reason why this work has just chosen those model resolutions.

Reply: Computation is a big burden for distributed modeling, computation in this article took more than a year as it needs a few tries to finish a complete run for one DEM resolution. Thanks the reviewer for a rapid comment, and in the past three months, the authors tried two more model resolutions at 400m and 600m, the results did not change the conclusion of this article based on the previous results. In this article, these new results has been added, please see the revised article, Fig. 6 and Fig. 7, Table 5 and Table 6.

5. In this paper, the abstract should be more concise and the motivation is not very clear.

Reply: The abstract has been rewritten, please see the revised article.

6. From line 116 to 133, the related works should be classified and then summarized concisely.

Reply: As these works have been published for some time, so to make this article concise, they are only mentioned in this paper, and will not be further summarized in detail, so revision is not done for this part.

7. If this work did not modify the Liuxihe Model published in the previous works, I suggest section 3.1 and 3.2 should be merged into one part. The description of the model could be reduced. Some contents in Section "introduction" and 3.1 are repeated.

Reply: The other reviewer suggested a more detailed introduction to the model. To combine the comments of both reviewers, considering Liuxihe Model employed in this study has been published in internationally refereed journals, so only a briefly introduction to the model structure, the components and algorithms used is kept, and move to the second section: Method and data. Please see the revised article.

8. The coordinate information in the maps of Fig. 1-3 and Fig. 6 should be displayed. The plotting scales should be the same for all maps.

Reply: The authors think there is no need to add the coordinate, so no revision be done.

9. The font in some figures should be accord with that in the manuscript. The units in some figures look unprofessional. For example, in Fig. 4(c), the title of the x axis should be "date" without unit.

Reply: Done in the revision.

10. The trend lines in Fig. 5 and 7 are not clear, especially during the flood. All the plots should be rearranged in the panels.

Reply: Done in the revision.

Please also note the supplement to this comment:
http://www.hydrol-earth-syst-sci-discuss.net/hess-2016-489/hess-2016-489-AC1-supplement.pdf

**Supplement:**

[revised manuscript text omitted]

Table 6 Optimized parameters with different model spatial resolution*

[revised manuscript text omitted]

---

## Author Comment (AC2) · 8 Dec 2016

Anonymous Referee # 2 This manuscript discusses an interesting topic of using a physically-based distributed model for flood forecasting in large watershed. It deals with challenges of computational speed and forecasting accuracy, possibly advancing flood mitigation and control decisions in practice. Following suggestions will help the authors improve the manuscript:

Reply: Thank the reviewer for his/her comments, and revisions will be down based on the reviewer's comments. Following are responses to the reviewer's comments one by one.

(1) Sections 3.2 _ 3.4 need to be reorganized by moving the results of the model (e.g., calibration and validation performances) to a subsection of "5. Results." These three sections can be combined into one subsection to just discuss the approaches for model calibration and validation.

Reply: The article has been reorganized based on the reviewer's comment. The revised article now as five parts, including introduction, method and data, results, discussions and conclusions. Please see the revised article.

(2) A section of Discussion is lacking. This section should be added to thoroughly discuss the major findings of this study and assess these findings in accordance with those reported by others. The purpose of this section is to put this study on the international arena.

Reply: A section of discussions has been added to address the reviewer's concerns, please see the revised article. Thank the reviewer for this comment, it improves the quality of this article.

(3) Both section Abstract and Conclusions are too long. They need to be rewritten to concisely highlight the importance/scientific contributions of the study, study approaches, and major findings. Some generic statements can be deleted.

Reply: The abstract has been rewritten more concisely, but the conclusion remains not changed, please see the revised article.

(4) Appropriate details of formalization should be provided to help audiences better understand your contributions. For example, the descriptions of the model core components, including saturation excess method, kinematic wave approximation, diffusive wave approximation, and particle swarm optimization (PSO) algorithm, are too simple. Some necessary formulas/equations should be presented because these methods could have distinctly different complexity levels, depending on how they are formulated.

Reply: The other reviewer suggested that the model introduction is simplified as the model employed is published elsewhere. To combine the comments of both reviewers, considering Liuxihe Model employed in this study has been published in internationally refereed journals, so only a briefly introduction to the model structure, the components and algorithms used is kept, and move to the second section: Method and data. Please see the revised article.

(5) More descriptions should be provided for the statistics (e.g., Nash-Sutclife coefficient and water balance coefficient). What are the thresholds of these statistics, above which the model can be judged to have a good performance?

Reply: As these formulas are well known and could be retrieved from many publications, so they are not added in the revised article.

(6) Some subject terminologies should be clarified. For example, the manuscript uses several different adjectives, such as "disastrous", "serious", "huge", and "large", to describe flood magnitude. How do you classify the floods, in terms life or economic losses?

Reply: These has be revised based on the reviewer's suggestion, please see the revised article.

(7) In Table 6 and other tables/figures, the units for some quantities are missing. Also, the values of Manning's n are larger than one. How can this be possible? In Table 6, you reported Manning's n = 1.17 _ 1.49.

Reply: Values in table 6 are adjusting coefficient of the optimized parameters to the initial parameters, so values of the final optimized parameters are initial parameters time adjusting coefficient. A footnote has been put under table 6 to avoid confusion.

(8) Lines 292-294: if observed data are not available, how can you optimize the model parameters? What to be optimized?

Reply: If observed data are not available, it is difficult to optimize the model parameters. But for a big watershed, it usually has some data, and Liuxihe Model could optimize model parameter with limited observation data, so the model parameters in large watershed could be optimized.
(9) The annotated manuscript has some specific comments. I suggest that the manuscript be proofread by an English native speaker. There are a number of awkward phrases/words and confusing sentences. Please also note the supplement to this comment: http://www.hydrol-earth-syst-sci-discuss.net/hess-2016-489/hess-2016-489-RC2-supplement.pdf

Reply: English has been improved based on the annotated manuscript with reviewer's comment by the authors, thanks the reviewer for taking time to make this comments, it helps to improve the English much. Proofread was not done by others.

Please also note the supplement to this comment:
http://www.hydrol-earth-syst-sci-discuss.net/hess-2016-489/hess-2016-489-AC2-supplement.pdf

**Supplement:**

[revised manuscript text omitted]

Table 6 Optimized parameters with different model spatial resolution*

[revised manuscript text omitted]